# An integrated personal and population-based Egyptian genome reference

Inken Wohlers[1], Axel Künstner[1], Matthias Munz[1], Michael Olbrich[1], Anke Fähnrich[1], Verónica Calonga-Solís[1,2], Caixia Ma[3], Misa Hirose[4], Shaaban El-Mosallamy[5], Mohamed Salama[5,6], Hauke Busch[1,7✉] & Saleh Ibrahim[4,7✉]

A small number of de novo assembled human genomes have been reported to date, and few have been complemented with population-based genetic variation, which is particularly important for North Africa, a region underrepresented in current genome-wide references. Here, we combine long- and short-read whole-genome sequencing data with recent assembly approaches into a de novo assembly of an Egyptian genome. The assembly demonstrates well-balanced quality metrics and is complemented with variant phasing via linked reads into haploblocks, which we associate with gene expression changes in blood. To construct an Egyptian genome reference, we identify genome-wide genetic variation within a cohort of 110 Egyptian individuals. We show that differences in allele frequencies and linkage disequilibrium between Egyptians and Europeans may compromise the transferability of European ancestry-based genetic disease risk and polygenic scores, substantiating the need for multi-ethnic genome references. Thus, the Egyptian genome reference will be a valuable resource for precision medicine.

[1] Medical Systems Biology Division, Lübeck Institute of Experimental Dermatology and Institute for Cardiogenetics, University of Lübeck, Ratzeburger Allee 160, 23562 Lübeck, Germany. [2] Department of Genetics, Federal University of Paraná (UFPR), Centro Politécnico, Jardim das Américas, 81531-990 Curitiba, Brazil. [3] Novogene (UK) Company Limited, 25 Cambridge Science Park, Milton Road, CB4 0FW Cambridge, UK. [4] Genetics Division, Lübeck Institute of Experimental Dermatology, University of Lübeck, Ratzeburger Allee 160, 23562 Lübeck, Germany. [5] Medical Experimental Research Center (MERC), Mansoura University, Elgomhouria St., Dakahlia Governorate 35516 Mansoura, Egypt. [6] Institute of Global Health and Human Ecology, The American University in Cairo, AUC avenue, 11835 Cairo, Egypt. [7] These authors contributed equally: Hauke Busch, Saleh Ibrahim. ✉email: Hauke.Busch@uni-luebeck.de; Saleh.Ibrahim@uni-luebeck.de

With the advent of personal genomics, population-based genetics as part of an individual's genome is indispensable for precision medicine. Currently, genomics-based precision medicine compares the patients' genetic make-up to a reference genome[1], a genome model inferred from individuals of mostly European descent, to detect risk mutations that are related to disease. However, genetic and epidemiologic studies have long recognized the importance of ancestral origin in conferring genetic risk for disease. Risk alleles and structural variants (SVs) can be missing from genetic reference data or can have different population frequencies, such that alternative pathways become disease related in patients of different ancestral origin, which motivates the establishment of national or international multi-ethnic genome projects[2]. At present, there are several population-based sequencing efforts that aim to map specific variants in the 100,000 genome projects in Asia[3] or England[4]. Furthermore, large-scale sequencing efforts currently explore population, society and history-specific genomic variations in individuals in Europe[5], North America[6], Asia[3] and, recently, the first sub-Saharan Africans[7,8]. Nonetheless, there is still little genetic data available for many regions of the world. In particular, North African individuals are not adequately represented in current genetic data sets, such as the 1000 Genomes[9], TOPMED[6] or gnomAD[10] databases. Consequently, imminent health disparities between different world populations have been noted repeatedly for a decade[11–14].

In recent years, several high-quality de novo human genome assemblies[15–18] and, more recently, pan-genomes[7] have extended human sequence information and improved the de facto reference genome GRCh38[1]. Nonetheless, it is still prohibitively expensive to obtain all-embracing genetic information, such as high-quality de novo assembled personal genomes for many individuals. Indeed, previous genetic studies assess only a subset of variants occurring in the Egyptian population, e.g., single nucleotide polymorphisms (SNPs) on genotyping arrays[19,20], variants in exonic regions via exome sequencing[21] or variants detectable by short-read sequencing[22,23].

Here, we present the Egyptian genome reference (Egyptref)—a comprehensive North African genetic reference data set—as an initial and essential step towards personal genomics-based precision medicine in this region. It is a combination of (i) a de novo assembled Egyptian personal genome together with phased variants and (ii) a population-genetic characterization of Egyptians based on 19,758,992 single nucleotide variants (SNVs), 121,141 SVs and mitochondrial haplogroups. Based on the Egyptian personal genome, we here report 1180 genes showing haplotypic expression in blood and 40 common unique insertions, two of which are likely novel. Based on the population genome, we report four major genetic ancestry components in Egyptians and 1198 Egyptian population-specific variants, 49 of which are novel. Finally, we find that haplotypes for disease risk loci identified in European cohorts differ from Egyptian haplotypes, which may impact genetic risk assessment. We anticipate that an Egyptian population genome reference will strengthen precision medicine efforts that eventually benefit nearly 100 million Egyptians, e.g., by providing allele frequencies (AFs) and linkage disequilibrium (LD) between variants, information that is necessary for both rare and common disease studies. Likewise, our genome will be of universal value for research purposes, since it contains both European and African variant features. Most genome-wide association studies (GWAS) are performed in Europeans[24], but genetic disease risk may differ, especially for individuals of African ancestry[25]. Consequently, Egyptref will be well suited to support recent efforts to include Africans in such genetic studies, for example, by serving as a benchmark data set for SNP array construction and variant imputation or for fine-mapping of disease loci.

## Results

**De novo assembly of the genome of an Egyptian individual.** Our Egyptian genome is based on a high-quality human de novo assembly for one male Egyptian individual (see workflow in Supplementary Fig. 1). This assembly was generated from PacBio, 10x Genomics and Illumina paired-end sequencing data at overall 270x genome coverage (Supplementary Data 1). For this personal genome, we constructed two draft assemblies, one based on long-read assembly by an established assembler, FALCON[26], and another based on the assembly by a novel assembler, WTDBG2[27], which has a much shorter run time with comparable accuracy (cf. Supplementary Fig. 1). Both assemblies were polished using short reads and further polishing tools. For the FALCON-based assembly, scaffolding was performed, whereas we found that the WTDBG2-based assembly was of comparable accuracy without scaffolding (Table 1). Sex chromosomal sequences have not been manually curated. The WTDBG2-based assembly was selected as the meta assembly basis, because it performs similarly or better than the FALCON-based assembly according to various quality control (QC) measures. The former did not require scaffolding, and thus there are no N bases or scaffolding errors. Overall, it has about 50% fewer disagreements to the reference. This holds true even when ignoring those disagreements in centromeres and in segmental duplications and after correction for SVs (Supplementary Data 2). Where larger gaps outside centromere regions occurred, we complemented this assembly with sequence from the FALCON-based assembly (Supplementary Data 3) to obtain a final Egyptian meta-assembly, denoted as EGYPT (for overall assembly strategy, see Supplementary Fig. 1). Both the base assemblies and the final meta assembly are of high quality and complementary and they are comparable to the publicly available assembly AK1 of a Korean individual[16] and the assembly of a Yoruba individual, here denoted YORUBA, (GenBank assembly accession GCA_001524155.4) in terms of genome length and various QC measures (Table 1, extended version in Supplementary Data 2). Assembly quality is confirmed by QC measures assessed by QUAST-LG[28] (Supplementary Data 2), NA-values (Supplementary Fig. 2), k-mer multiplicity with KAT[29] (Supplementary Figs. 3–5), QV values of more than 40 and by dot plots of alignment with reference GRCh38 (Supplementary Figs. 6–10).

When annotating repeats with REPEATMASKER, about 49% of the genome were identified as repetitive with its contents similarly identified as in AK1 or YORUBA (Supplementary Data 4).

The meta-assembly was complemented with high-quality phasing information (Supplementary Data 5). EGYPT SNVs and small insertions and deletions (indels) called using short-read sequencing data were phased using high-coverage 10x linked-read sequencing data. This resulted in 3,834,900 of 4,008,080 autosomal variants being phased (95.7%). Furthermore, nearly all (99.41%) of the genes with lengths <100 kb and more than one heterozygous SNV were phased into a single phase block. We identified 22 runs of homozygosity (ROH) (Supplementary Data 6), out of which 16 are larger than 5 Mb and sum up to overall 192 Mb, which indicates consanguinity at the level of parental third-degree relationship[30].

**Genetic variation among Egyptians.** Based on the personal Egyptian genome, we constructed an Egyptian population genome by considering genome-wide SNV AFs in 109 additional Egyptians (Supplementary Data 7). This approach enabled the characterization of the major allele (i.e., the allele with highest AF) in the given Egyptian cohort. To accomplish

**Table 1 Comparison of main assembly characteristics and quality metrics.**

|  | EGYPT | EGYPT_wtdbg2 | EGYPT_falcon | AK1 | YORUBA |
|---|---|---|---|---|---|
| Assembly level | meta | contig | scaffold | scaffold | chromosome |
| Effective genome size | 2,820,489,739 | 2,733,934,177 | 2,897,551,797 | NA | NA |
| # Sequences | 3235 | 3106 | 1615 | 2832 | 1647 |
| Longest sequence | 88,566,048 | 88,566,048 | 84,324,762 | 113,921,103 | 248,986,603 |
| # N's per 100 kbp | 0 | 0 | 209.01 | 1285.7 | 7180.2 |
| Base level QV | 42.4 | 42.9 | 43.0 | 50.4[a] | NA |
| # Genes (thereof # partial) | 20,908 (3226) | 20,613 (3229) | 21,176 (1578) | 21,047 (1396) | 21,077 (1721) |
| Genome fraction w.r.t. GRCh38 (%) | 94.174 | 92.247 | 95.924 | 95.177 | 95.391 |
| Duplication ratio w.r.t. GRCh38 | 1.01 | 0.999 | 1.018 | 1.023 | 1.088 |
| Largest GRCh38 alignment | 75,492,126 | 75,492,126 | 56,458,009 | 58,219,133 | 65,512,502 |
| Total GRCh38 aligned length | 2,800,100,449 | 2,713,712,375 | 2,865,356,241 | 2,829,006,639 | 2,832,740,986 |
| NG50 w.r.t. GRCh38 | 20,857,787 | 20,857,787 | 28,071,354 | 39,609,866 | 145,208,384 |
| LG50 w.r.t. GRCh38 | 35 | 35 | 33 | 24 | 9 |
| NGA50 w.r.t. GRCh38 | 11,187,777 | 11,187,777 | 8,226,500 | 13,028,687 | 19,529,238 |
| LGA50 w.r.t. GRCh38 | 71 | 71 | 95 | 66 | 43 |
| # GRCh38 differences >1 kb (thereof # outside centromeres) | 1276 (1103) | 1276 (1103) | 3499 (2832) | 1952 (1685) | 1756 (1472) |
| # GRCh38 mismatches per 100 kbp | 139 | 138.72 | 143.64 | 126.92 | 141.56 |
| # GRCh38 indels per 100 kbp | 32.09 | 31.74 | 40.06 | 32.77 | 46.95 |
| K-mer-based compl. w.r.t. GRCh38 (%) | 86.01 | 85.15 | 87.75 | 87.68 | 85.82 |

The table lists the final EGYPT meta assembly, the two alternative base assemblies EGYPT_wtdbg2 and EGYPT_falcon and two publicly available assemblies of the genomes of a Korean (AK1) and Yoruba individual (YORUBA). Metrics were largely obtained with QUAST-LG. The complete QUAST-LG report and additional assembly metrics are provided in Supplementary Data 2.
[a]Based on error rate estimated by AK1 authors Seo et al.[16].

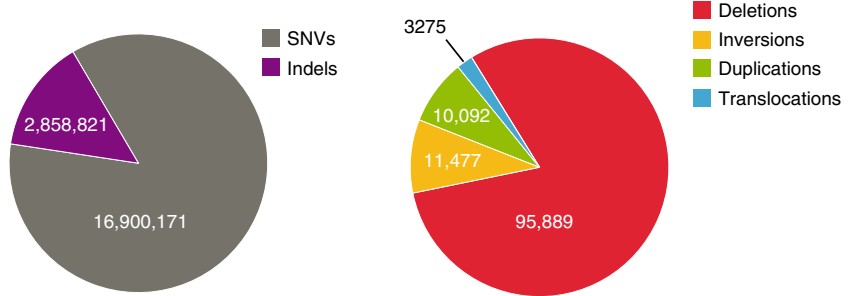

**Fig. 1 Number of various genetic variant types identified in the Egyptian cohort.** Left: The number of single nucleotide variants (SNVs) and small insertions and deletions (indels). Right: The number of structural variant (SV) calls: deletions, inversions, duplications and translocations. In addition, 408 insertions have been called.

this, we called variants using short-read data of 13 Egyptians sequenced at high coverage and 97 Egyptians sequenced at low coverage. Although sequence coverage affects variant-based statistics (Supplementary Fig. 11), due to combined genotyping, most variants could also be called reliably in low coverage samples (Supplementary Fig. 12). We called a total of 19,758,992 SNVs and small indels (Supplementary Fig. 13) in all 110 Egyptian individuals (Fig. 1). The number of called variants per individual varied between 2,901,883 and 3,934,367 and was correlated with sequencing depth (see Supplementary Figs. 11 and 12). This relationship was particularly pronounced for low coverage samples. The majority of variants were intergenic (53.5%) or intronic (37.2%) (Supplementary Fig. 14). Only ~0.7% of the variants were located within coding exons, of which 54.4% were non-synonymous and thus cause a change in protein sequence and, possibly, structure (Supplementary Fig. 15).

Using short-read sequencing data of 110 Egyptians, we called 121,141 SVs, most of which were deletions, but also included inversions, duplications, insertions and translocations of various orders of magnitude (Fig. 1, Supplementary Figs. 16 and 17).

Similar to SNVs, the number of SV calls also varied between individuals (Supplementary Fig. 18) and is slightly affected by coverage (Supplementary Fig. 19). After merging overlapping SV calls, we obtained an average of 2,773 SVs per Egyptian individual (Supplementary Data 8, Supplementary Figs. 20–22).

Using the EGYPT de novo assembly, we searched for unique insertions that are common in Egyptians. Towards this, we took a similar approach as Wong et al.[31] and mapped all short-read data against the GRCh38 reference genome including other decoy or alternative haplotype sequences using the GATK[32] bundle. Altogether we identified 40 unique insertions longer than 500 bp with a total length of 40 kb, for which we required for every base in the identified region to have a minimal coverage of 5 reads in at least 10 Egyptian individuals (Supplementary Data 9). Of these sequences, 28 have been mentioned before by Wong et al.[31], and 10 more in different studies within the last 15 years[33–36]. Two out of the 40 insertions are most likely novel. In addition, one region contains three unique insertions, of which two contain additional, novel sequences longer than 500 bases. Closer inspection reveals that these sequences are located within a region of two 50 kb gaps (i.e., N sequences) in the GRCh38 reference genome at chromosome loci

chr13:111,703,856-111,753,855 and chr13:111,793,442-111,843,441 with about 40 kb of reference sequence between the gaps. The EGYPT, AK1, and YORUBA assembly sequences that cover this 140 kb reference sequence from chr13:111,703,856 to 111,843,441 are very similar (Supplementary Figs. 23–25). They all align about 4 kb from the 40 kb reference sequence between the gaps, only, but at the very beginning of the respective assembly sequence (Supplementary Figs. 26–28). Performing a BLAST search of the 140 kb EGYPT assembly sequence reveals an overall 44 kb alignment in five, mainly consecutive, large alignment blocks to Homo sapiens chromosome 13 clone WI2-2182D8 (AC188786.1) from position 1 to 44,382, see Supplementary Fig. 29. This large reference genome region that contains the largest gap covering sequence reported for AK1[16] is not resolved yet.

Overall, we identified 330 SNVs and indels in 36 of 40 non-reference sequences (Supplementary Data 10). The percentage of reads that could not be mapped to GRCh38 or GATK bundle sequences, but which were mappable against the de novo assembly is on average 8.6%, but for some individuals up to 34.2% (cv. Supplementary Fig. 30). Previously unmapped short reads of 110 Egyptians covered positions for more than 19 Mb of the Egyptian de novo assembly. Unique sequences that are commonly shared among Egyptians illustrate that additional reference genomes are needed to capture genetic diversity that is neither assessable by short read sequencing nor with the current human reference genome.

In addition, the large number of assembly positions to which such short reads map which could not be mapped to the reference genome GRCh38 (including widely used supplementary sequences included in the GATK bundle), indicate a need for further assembly-based reference data and for new approaches to better capture genetic diversity.

**Population genetic characterization**. Genotype principal component analysis (PCA) of the Egyptian cohort shows a homogeneous group for which the assembly individual is representative (Supplementary Figs. 31–37).

We genetically characterized the Egyptian population with respect to 143 other populations of the world using variant data of 5429 individuals in total. For this, we combined five different data sets: (1) a recently published whole-genome sequencing (WGS)-based variant data set from 929 individuals of the Human Genome Diversity Project (HGDP), covering 51 populations[37]; (2) 2504 individuals from 26 populations of the 1000 Genomes project for which phase 3 genotypes are available[9]; (3) WGS-based variant data from 108 Qatari individuals[38]; (4) SNP array-based variant data of 478 individuals from five countries of the Arabian Peninsula[20]; (5) 1305 individuals from 68 African, European, Western and Southern Asian populations that were compiled from eight different publications into a recent SNP array-based variant data set[39]. All individuals and their annotations are provided in Supplementary Data 11, data sources are described in Supplementary Data 12. A PCA of the data shows a genetic continuum between Europeans, Africans, East Asians and Americans along the first three principal components, see the interactive HTML-based Figure in Supplementary Data 13. Egyptians are located on the European-African axis and close to Europeans. Their genetic variance spreads to a small degree in the direction of the Asian axis, akin to further individuals from the Middle East (see Fig. 2c). To preclude a technical bias when intersecting WGS with SNP array data, we compared the analysis results when using whole-genome data, only, or when intersecting WGS data with SNP array data of 398 Egyptians (cf. Supplementary Data 12) and found comparable results in both cases (Supplementary Fig. 38). The Egyptian PCA location is

further supported by an admixture analysis. Our analysis specifies $k = 24$ as the optimal number of genetic components for the entire data set, i.e., having the smallest cross validation error (see Supplementary Fig. 39 for results for $k = 10$ to $k = 25$). Accordingly, the genetics of Egyptian individuals comprises four distinct ancestry components that sum up to 75% on average. Egyptians have a Middle Eastern, a European/Eurasian, a North African and an East African component with 27%, 24%, 15% and 9% relative influence, respectively (Fig. 2a). According to our cohort, Egyptians show genetically little heterogeneity, with little variance in the proportion of ancestry components between the individuals (Supplementary Figs 40 and 41). With a focus on populations from the Horn of Africa, the four components we identified have been described before by Hodgeson et al.[40] in a cohort of 2194 individuals from 81 populations (mainly 1000 Genomes and HGDP) and substantially fewer variants ($n = 16,766$), but including also 31 Egyptians. They and others hypothesize that most non-African ancestry, i.e., the Eurasian/European and Middle Eastern components in the populations from North Africa and the Horn of Africa is resulting from prehistoric back-to-Africa migration[19,40]. Recently, Serra-Vidal et al.[41] describe North Africa as a melting pot of genetic components, attributing most genetic variation in the region also to prehistoric times. Here, we confirm previously identified genetic components, yet using 2.5 times as many individuals, and using WGS data for the majority of them. This is thus the hitherto most comprehensive data set on genetic diversity world-wide and in this region.

The European, African, and Asian ancestry components of Egyptians are further supported by mitochondrial haplogroup assessment from mtDNA sequencing of 217 individuals in addition to haplogroups of the same 110 Egyptian individuals described earlier, for which 100 haplogroups are available from the literature[22]. mtDNA sequencing revealed that Egyptians have haplogroups most frequently found in Europeans (e.g., H, V, T, J, etc.; >60%), Africans (e.g., L with 24.8%) or Asians/East Asians (e.g., M with 6.7%) (Supplementary Fig. 42). Overall, this supports the admixture and PCA analysis and the notion that Egypt's transcontinental geographical location shaped Egyptian genetics.

Lastly, we characterized the Egyptian population with respect to ROH. The distribution of overall length of ROHs larger than 5 Mb is comparable for the Egyptian population and Middle Eastern populations and, to lesser extent, also for other North African and Western Asian populations. In comparison, Europeans and Sub-Saharan Africans have usually shorter ROHs, see Fig. 2b. Abundance of long ROHs is typical for the Greater Middle East[21] and reflects the common practice of consanguineous marriages in this region.

In total, we identified 6,599,037 common Egyptian SNVs (minor allele frequency (MAF) > 5%, genotypes in a minimum of 100 individuals), of which 1198 are population-specific; i.e., they are either rare (MAF < 1%) or not detected in any other population in the 1000 Genomes[9], gnomAD database[10] or TOPMed[6] (Supplementary Data 14). These numbers are comparable to population-specific variant numbers reported previously for 1000 Genomes populations[42]. Four SNVs likely have a molecular impact (Supplementary Data 15), indicated by a CADD[43] deleteriousness score >20. SNP rs143563851 (CADD 24.2) has recently been identified in 1% of individuals of a cohort of 211 Palestinians in a study that performed targeted sequencing of blood group antigen synthase GBGT1[44]. SNP rs143614333 (missense variant in gene CR2, CADD 23.6) is in ClinVar[45], with three submitters reporting that the variant is of uncertain clinical significance. In addition, we obtained 49 variants with no dbSNP[46] rsID (Supplementary Data 16). These numbers of

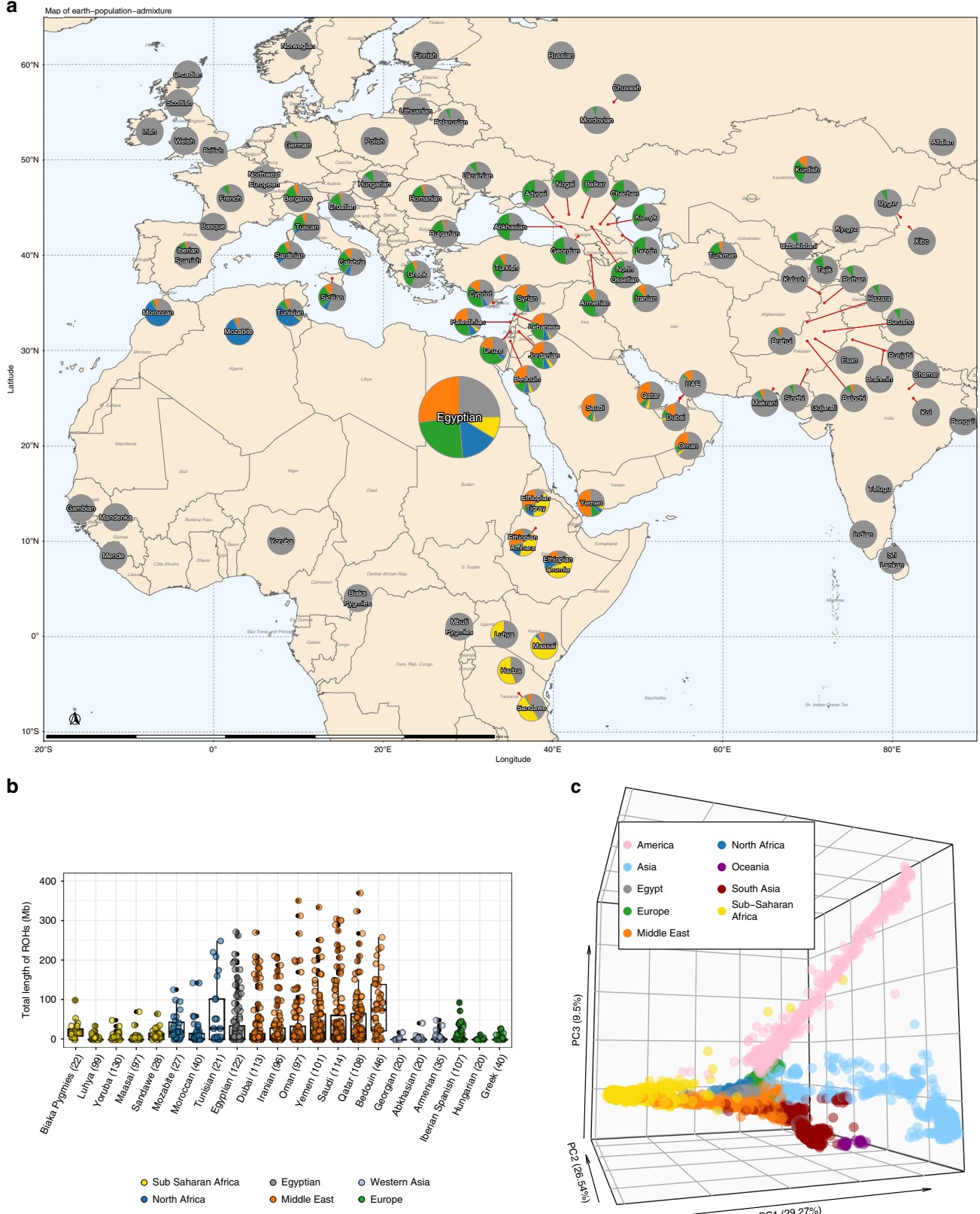

**Fig. 2 Population genetic characterization of the Egyptian population. a** The four largest ancestry components in the Egyptian population for African, European and Western Asian populations. Colors denote the world region in which the respective component is most prevalent – Orange: Middle East, Green: Eurasia; Blue: North Africa; Yellow: Sub-Saharan Africa. **b** Box plots for total length of runs of homozygosity (ROH) for the Egyptians and several populations from relevant world regions. Colors denote world regions. The number n of individuals per population is given in brackets. One Qatari individual with a total length of ROH longer than 400Mb is not shown. Box plots display median and lower/upper quartiles; whiskers denote the most extreme data point no more than 1.5 times the interquartile range; outliers are data points extending beyond whiskers. **c** Principal component analysis with $n = 5429$ individuals from populations world-wide. The dots are colored according to world region.

population-specific SNPs, of which some are likely to have an immediate impact on clinical characteristics and diagnostics, indicate insufficient coverage of the genetic diversity of the world's population for precision medicine and thus the need for local genome references. To detect a putative genetic contribution of Egyptian population-specific SNPs towards molecular pathways, phenotypes or disease, we performed gene set enrichment analysis for all 461 protein-coding genes that were annotated to population-specific SNPs by Ensembl VEP[47]. ENRICHR, a gene list enrichment tool incorporating 153 gene sets and pathway databases[48], reports that genes from obesity-related traits of the GWAS catalog 2019 collection are over-represented (adj. p-value: 1.02E-6; 49 of 804 genes), which might hint at population-specific metabolism regulation that is linked to body weight.

**Haplotypic expression**. Variants that are not protein coding may have a regulatory effect that affects gene and eventually protein expression. Using blood expression data obtained from RNA sequencing for the EGYPT assembly individual in conjunction with 10x Genomics sequencing-based phased variant data, we identified genes with haplotype-dependent expression patterns (see Supplementary Fig. 43 for the analysis overview and Supplementary Figs. 44 and 45 for the results). We report 1180 such genes (Supplementary Data 17). Of these, variants contained in haplotypes of 683 genes (58%) have previously reported expression quantitative trait loci (eQTLs) in blood according to Qtlizer[49], for 380 genes supported by multiple studies. For 370 genes (31%), the strongest associated blood eQTL SNV is haplotypically expressed, and for 131 genes, the best eQTL has been previously reported by multiple studies. Concordance of haplotypic expression with eQTLs indicates that a common variant may affect gene expression; discordance hints towards a rare variant.

**Egyptian ancestry likely impacts genetic risk assessment**. We investigated the impact of Egyptian ancestry on disease risk assessment by integrating Egyptian variant data with the GWAS catalog[50], a curated database of GWAS. According to the GWAS catalog, most published GWAS are performed on Europeans[24], and only a single study has been performed on Egyptians[51] (by one of the co-authors). Furthermore, only 2% of individuals included in GWAS are of African ancestry[24]. AFs, LD and genetic architecture can differ between populations, such that results from European GWAS cannot necessarily be transferred[25]. This lack of transferability also compromises the prediction of an individual's traits and disease risk using polygenic scores: such scores are estimated to be approximately one-third as informative in African individuals compared to Europeans[52]. From the GWAS catalog, we constructed a set of 4008 different, replicated, high-quality tag SNPs (i.e., one strongest associated SNP per locus) from European ancestry GWAS for 584 traits and diseases. We compared the tag SNPs' AFs and proxy SNVs in the Egyptian cohort ($n = 110$) and Europeans from 1000 Genomes ($n = 503$) (Supplementary Data 18). Egyptian AFs of tag SNPs are comparable to European AFs, with a tendency to be lower (Fig. 3a). There are variants common in Europeans (AF > 5%) but rare in Egyptians (AF < 5%) (Supplementary Fig. 46). A total of 261 tag SNPs are not present in the Egyptian cohort (~7%), clearly indicating a need to perform GWAS in non-European populations using genotyping techniques that better capture population-genetic diversity to further elucidate disease risk conferred by these loci. We investigated differences in LD structure using an approach that is used for fine-mapping of GWAS data, which identifies proxy variants (illustrated in Fig. 3b). Proxy variants are variants correlated with the tag GWAS SNP, i.e., in high LD (here, $R^2 > 0.8$). The post-GWAS challenge is the identification of

a causal variant from a set of variants in LD (tag SNP and proxy variants). We found that the number of proxy variants was much lower in the Egyptian cohort (Fig. 3c), likely due to shorter haplotype blocks known from African populations. This indicates that LD differences between Egyptians and Europeans may compromise GWAS transferability and European ancestry-based polygenic scores. However, Egyptian proxy variants are usually included in the larger set of European proxy variants (Fig. 3d). An example is variant rs2075650 (a locus sometimes attributed to gene *TOMM40*), which has been linked to Alzheimer's disease in seven GWASs (cf. Supplementary Fig. 47). This tag SNP has seven proxy variants in Europeans but only two proxy variants in Egyptians. One European proxy, rs72352238, has also been reported as a GWAS tag SNP, but it is not a proxy of rs2075650 in Egyptians and may thus fail replication and transfer of GWAS results from the European to the Egyptian population.

## Discussion

With Egyptref, it will be possible to perform comprehensive integrated genome and transcriptome comparisons for Egyptian individuals in the future. This will shed light on personal as well as population-wide common genetic variants. As an example for personalized medicine for Egyptian-specific genetics we visualize the complete genetic information of the DNA repair-associated gene *BRCA2* from our study in the integrative genomics viewer[53] (IGV) and the variant phasing information within the 10x Genomics browser LOUPE in Fig. 4 and Supplementary Fig. 48, respectively. *BRCA2* is linked to the progression and treatment of breast cancer and other cancer types[54], if mutated. The IGV depicts the sample coverage based on sequencing data from PacBio, 10x Genomics and Illumina (whole genome as well as RNA) for the personal EGYPT genome together with common Egyptian SNPs. Variants previously assessed in a breast cancer GWAS[54] are displayed as Manhattan plot; note the three significant GWAS SNPs between positions 32,390 and 32,400 kb. The bottom compares the identified SNVs and indels from the Korean and Yoruba de novo assembly with our de novo EGYPT assembly. Visual inspection of both small and structural variations at the personal and population-based genome levels already yields significantly different variants, which might be important for genetic counseling and detection of inherited risks for cancer.

In conclusion, we constructed the first Egyptian—and North African—genome reference, which is an essential step towards a comprehensive, genome-wide knowledge base of the world's genetic variations. The wealth of information it provides can be immediately utilized to study in-depth personal genomics and common Egyptian genetics and its impact on molecular phenotypes and disease. This reference will pave the way towards a better understanding of the Egyptian, African and global genomic landscape for precision medicine.

## Methods

**Ethics statement**. This study was approved by the Mansoura Faculty of Medicine Institutional Review Board (MFM-IRB) Approval Number RP/15.06.62. All subjects gave written informed consent in accordance with the Declaration of Helsinki. This study and its results are in accordance with the Jena Declaration (https://www.uni-jena.de/unijenamedia/Universit%C3%A4t/Abteilung+Hochschulkommunikation/Presse/Jenaer+Erkl%C3%A4rung/Jenaer_Erklaerung_EN.pdf).

**Sample acquisition**. Samples were acquired from 10 adult Egyptian individuals. These were recruited from healthy relatives escorting patients admitted to Mansoura University hospital. Those consented to join the study and who met the main selection criteria, i.e., they identified as Egyptians up to the third generation, were included. Medical history was taken to ascertain no history of chronic diseases, followed by a full clinical examination by a medical doctor alongside routine laboratory investigations (Liver and kidney function tests and complete blood count (CBC)). For nine individuals, high-coverage Illumina short-read data were generated. For the assembly individual, high-coverage short-read data were

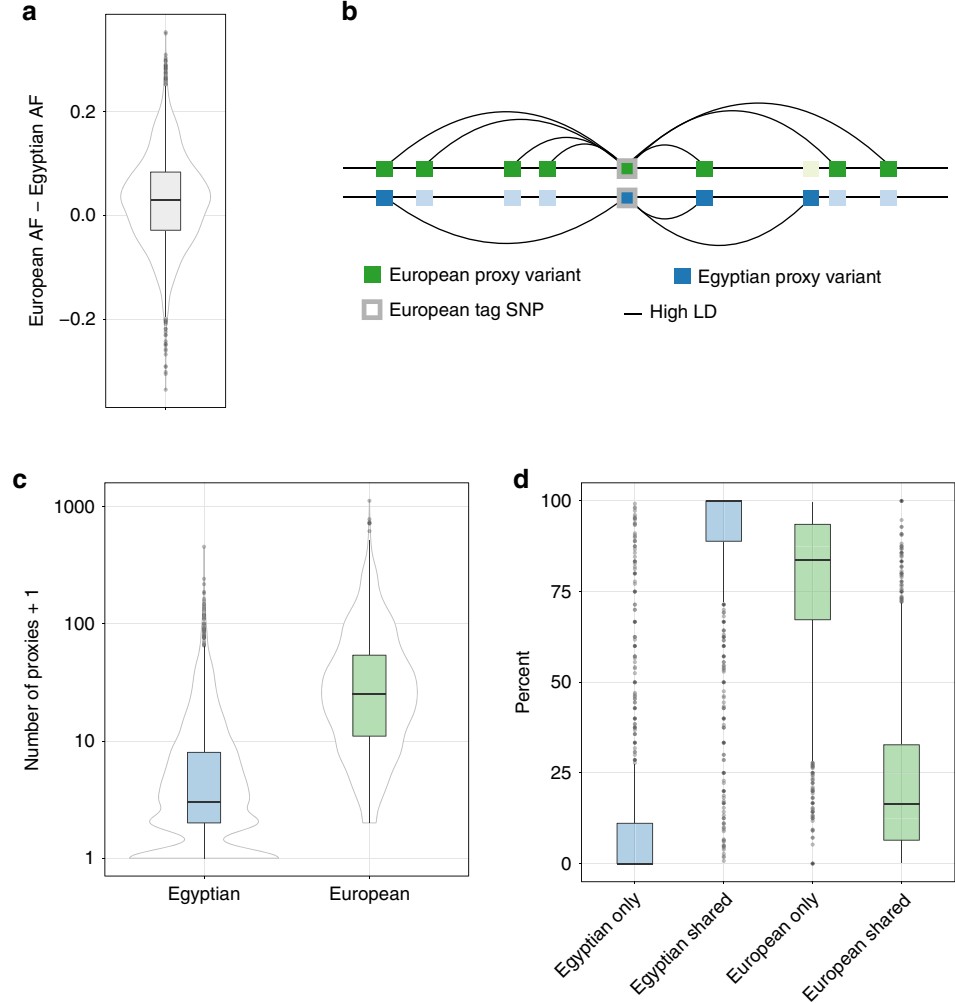

**Fig. 3 Comparison of Egyptian (blue) and European (green) allele frequencies (AF) and proxy variants for $n = 3,698$ GWAS tag SNPs called in a minimum of 100 Egyptians.** The box/violin plots display median and lower/upper quartiles; whiskers denote the most extreme data point no more than 1.5 times the interquartile range; outliers are data points extending beyond whiskers. The thin gray lines depict the respective distribution functions. **a** Box plot of AF differences. **b** Illustration of proxy variant counting shown in (**c**) and of proxy variant comparison shown in (**d**). A European GWAS tag SNP (center) and variants in Europeans (top) and Egyptians (bottom). Lines denote variants in high LD. The tag SNP has 7 proxy variants in Europeans and 3 in Egyptians. Light green/blue variants are no proxy variants in Europeans/Egyptians. Two proxy variants are shared. Thus 2 of 7 European (~29%) and 2 of 3 Egyptian (~67%) variants are shared. Further 5 of 7 European proxies are European-only (~71%) and 1/3 Egyptian proxies are Egyptian-only (~33%). **c** Box/violin plots depicting the number of proxies. **d** Box plots showing the percentage of shared and Egyptian/European-only proxy variants using the same $n = 3,698$ GWAS tag SNPs as in (**a**) and (**c**). European shared: Percentage of European proxy variants shared with Egyptian proxy SNPs. European only: Percentage of European proxy variants not shared with Egyptian proxies. Egyptian shared/Egyptian only respectively.

generated as well as high-coverage PacBio data and 10x Genomics data. Furthermore, we used public Illumina short-read data from 100 Egyptian individuals from Pagani et al.[22]. See Supplementary Data 1 and 7 for an overview of the individuals and the corresponding raw and result data generated in this study.

**PacBio data generation**. For PacBio library preparation, the SMRTbell DNA libraries were constructed following the manufacturer's instructions (Pacific Bioscience, www.pacb.com). The SMRTbell DNA libraries were sequenced on the PacBio Sequel and generated 298.2 Gb of data.

Sequencing data from five PacBio libraries were generated at overall 99x genome coverage.

**Illumina short-read data generation**. For 350 bp library construction, the genomic DNA was sheared, and fragments with sizes of ~350 bp were purified from agarose gels. The fragments were ligated to adaptors and amplified using PCR. The generated libraries were then sequenced on the Illumina HiSeq X Ten using PE150 and generated 312.8 Gb of data.

For the assembly individual, sequencing data from five libraries were generated at overall 90x genome coverage. For nine additional individuals, one library each was generated, amounting to an overall 305x coverage of sequencing data. For the 100 individuals of Pagani et al.[22], three were sequenced at high coverage (30×) and

97 at low coverage (8×). The average coverage over SNV positions for all 110 samples is provided in Supplementary Data 7.

**RNA sequencing data generation**. For RNA sequencing, ribosomal RNA was removed from total RNA, double-stranded cDNA was synthesized, and then adaptors were ligated. The second strand of cDNA was then degraded to generate a directional library. The generated libraries with insert sizes of 250–300 bp were selected and amplified and then sequenced on the Illumina HiSeq using PE150.

Overall, 64,875,631 150 bp paired-end sequencing reads were generated.

**10x Genomics sequencing data generation**. For 10x Genomics sequencing, the Chromium Controller was used for DNA indexing and barcoding according to the manufacturer's instructions (10x Genomics, www.10xgenomics.com). The generated fragments were sheared, and then adaptors were ligated. The generated libraries were sequenced on the Illumina HiSeq X Ten using PE150 and generated 272.7 Gb of data. Sequencing data from four 10x libraries was generated at overall 80x genome coverage.

**Construction of draft de novo assemblies and meta-assembly**. We used WTDBG2[27] for human de novo assembly followed by its accompanying polishing

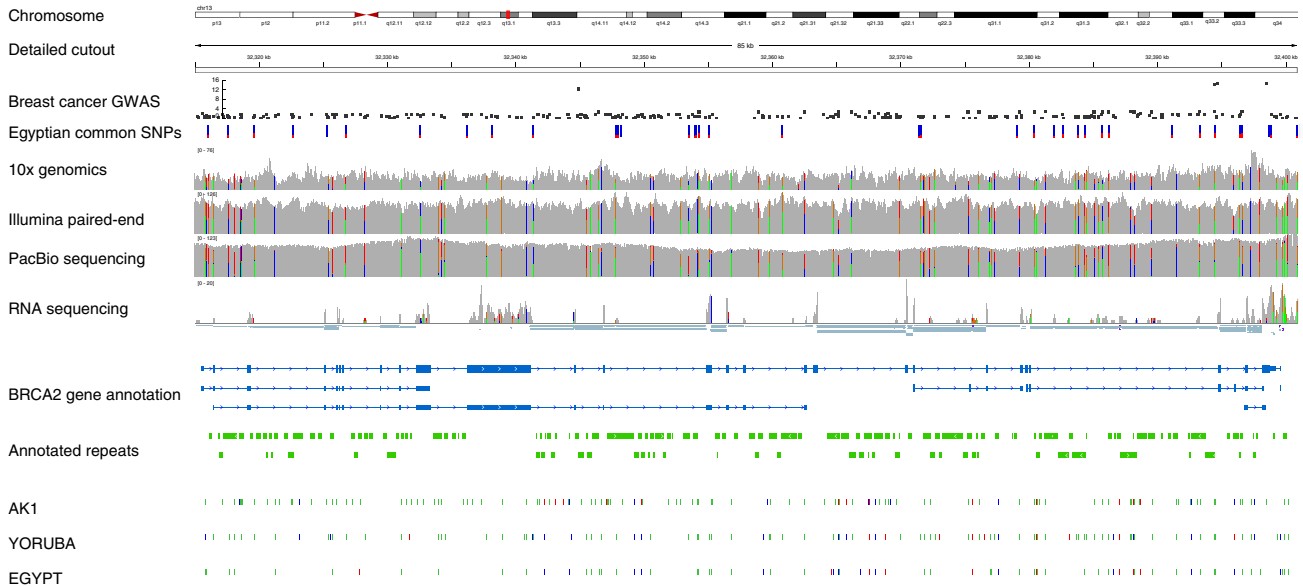

| Chromosome |
| Detailed cutout |
| Breast cancer GWAS |
| Egyptian common SNPs |
| 10x genomics |
| Illumina paired-end |
| PacBio sequencing |
| RNA sequencing |
| BRCA2 gene annotation |
| Annotated repeats |
| AK1 |
| YORUBA |
| EGYPT |

**Fig. 4 Integrative view of Egyptian genome reference data for the gene BRCA2, which is associated with breast cancer.** The rows denote from top to bottom: Genome location on chromosome 13 of the magnified region for BRCA2 (first and second row); GWAS data for breast cancer risk[54]; Variants that are common in the cohort of 110 Egyptians; Read coverage of genetic region based on 10x Genomics, Illumina paired-end and PacBio sequencing data; Coverage and reads of RNA sequencing data; BRCA2 gene annotation from Ensembl; Repeats annotated by REPEATMASKER; SNVs and indels identified by comparison of assemblies AK1, YORUBA and EGYPT with GRCh38. The colors denote base substitutions (green), deletions (blue) and insertions (red). The region above is included in a larger phase block chr13:25,831,216-33,523,430 and the corresponding variant phasing for the EGYPT individual is displayed in Supplementary Fig. 48.

tool WTPOA-CNS with PacBio reads and in a subsequent polishing run with Illumina short reads. This assembly was further polished using PILON[55] with short-read data (cf. Supplementary Methods: WTDBG2-based assembly).

An alternative assembly was generated by using FALCON, QUIVER[56], SSPACE-LONGREAD[57], PBJELLY[58], FRAGSCAFF[59], and PILON[55] (cf. Supplementary Methods: FALCON-based assembly).

Proceeding from the WTDBG2-based assembly, we constructed a meta-assembly. Regions larger than 800 kb that were not covered by this base assembly and were not located within centromere regions were extracted from the alternative FALCON-based assembly (Supplementary Data 3). See Supplementary Fig. 1 for an overview of our assembly strategy, including meta-assembly construction (cf. Supplementary Methods: Meta-assembly construction).

Assembly quality and characteristics were assessed with QUAST-LG[28]. In addition, we removed misassemblies in centromeres or in segmental duplication regions from the QUAST-LG report and furthermore removed SVs from misassemblies (cf. Supplementary Methods: Assembly comparison and QC). The extraction of coordinates for meta-assembly construction was performed using QUAST-LG output. K-mer multiplicity was assessed with KAT[29]. Following Porubsky et al.[60], we computed QV as $(-10)$ times log10 of the number of homozygous variants divided by the effective genome size. Towards this, we mapped all short reads to the assembly using BWA MEM and perform variant calling using FREEBAYES with default parameters. We kept only homozygous variants with a minimum quality of 10 using VCFTOOLS. Single-nucleotide differences were counted as difference of 1 bp, indel differences as the length differences between reference and alternative allele. Based on SAMTOOLS command stats, we computed the sum of bases with short read coverage as effective genome size.

**Repeatmasking.** Repeatmasking was performed by using REPEATMASKER[61] with RepBase version 3.0 (Repeatmasker Edition 20181026) and Dfam_consensus (http://www.dfam-consensus.org) (cf. Supplementary Methods: Repeat annotation).

**Unique inserted sequences.** We trimmed Illumina short sequencing reads of 110 Egyptian individuals using FASTP 0.20.0 with default parameters, mapped the output reads to GRCh38 and GATK bundle sequences using BWA 0.7.15-r1140 and sorted by chromosomal position using SAMTOOLS 1.3.1. Subsequently, we extracted reads that did not map to GRCh38 using SAMTOOLS with parameter F13 (i.e., read paired, read unmapped, mate unmapped) and repeated the mapping and sorting using the Egyptian de novo assembly. We merged the read-group specific BAM files for each sample and calculated the per base read depth using SAMTOOLS. Afterwards, we aggregated the results via custom scripts and extracted uniquely inserted sequences from the Egyptian de novo assembly. Insertions were defined as contiguous regions of at least 500 bp having a coverage of more than 5 reads per base in 10 or more samples.

Lastly, we BLASTed the obtained sequences against the standard databases (option nt) for highly similar sequences (option megablast) using a custom script. For the uniquely inserted sequences that we identified, we created a pileup over all BAM files containing the reads that did not map to GRCh38 using SAMTOOLS. Based on these pileups, we then called the variants using BCFTOOLS. Variants with quality of more than 10 were kept.

**Phasing.** Phasing was performed for the assembly individual's SNVs and short indels obtained from combined genotyping with the other Egyptian individuals, i.e., based on short-read data. These variants were phased using 10x Genomics data and the 10x Genomics LONGRANGER WGS pipeline with four 10x libraries provided for one combined phasing. See Supplementary Methods: Variant phasing for details.

**SNVs and small indels.** Calling of SNVs and small indels was performed with GATK 3.8[32] using the parameters of the best practice workflow. Reads in each read group were trimmed using Trimmomatic[62] and subsequently mapped against reference genome hg38 using BWA-MEM[63] version 0.7.17. Then, the alignments for all read groups were merged sample-wise and marked for duplicates. After the base recalibration, we performed variant calling using HaplotypeCaller to obtain GVCF files. These files were input into GenotypeGVCFs to perform joint geno-typing. Finally, the variants in the outputted VCF file were recalibrated, and only those variants that were flagged as PASS were kept for further analyses. We used FastQC[64], Picard Tools[65] and verifyBamId[66] for QC (cf. Supplementary Methods: Small variant QC).

**Variant annotation.** Variant annotation was performed using ANNOVAR[67] and VEP[47] (cf. Supplementary Methods: Small variant annotation).

**Structural variants.** SVs were called using DELLY2[68] with default parameters according to instructions on the DELLY2 website for germline SV calling (https://github.com/dellytools/delly) (cf. Supplementary Methods: Structural variant QC). Overlapping SV calls in the same individual were collapsed by the use of custom scripts. See Supplementary Methods: Collapsing SVs for details.

**Population genetics.** For population-genetic analyses, we compared the Egyptian variant data with variant data from five additional sources specified in Supplementary Data 12. Individuals together with their annotations are listed in Supplementary Data 11. Variant data were merged to contain only variants present in all data sets and subsequently filtered and LD pruned. Genotype PCA was computed using SMARTPCA[69] from the EIGENSOFT package. Admixture was computed with ADMIXTURE[70] (cf. Supplementary Methods: Population genetics and Supplementary Methods: SNP array-based Egyptian variant data). ROH were

computed on the same files that were used for PC computation and admixture using PLINK –homozyg. ROHs with size larger than 5 Mb were summed to obtain overall length of ROHs per individual.

**Mitochondrial haplogroups**. Haplogroup assignment was performed for 227 individuals using HAPLOGREP2[71]. Furthermore, mitochondrial haplogroups were obtained from Pagani et al.[22] for 100 individuals. See Supplementary Methods: Mitochondrial haplogroups for details.

**Population-specific variants**. Our set of common Egyptian SNVs comprises variants with genotypes in a minimum of 100 individuals whose alternative allele has a frequency of more than 5%. Those common Egyptian SNVs that are otherwise rare, i.e., have an AF of <1% in the 1000 Genomes, and gnomAD populations as well as in TOPMed were considered Egyptian-specific. AFs were annotated using the Ensembl API. Furthermore, a list of Egyptian common variants without dbSNP rsID was compiled, see Supplementary Methods: Small variant annotation for details.

**Haplotypic expression analysis**. RNA sequencing reads were mapped and quantified using STAR (Version 2.6.1.c)[72]. Haplotypic expression analysis was performed by using PHASER and PHASER GENE AE (version 1.1.1)[73] with Ensembl version 95 annotation on the 10x-phased haplotypes using default parameters. See Supplementary Methods: Haplotypic expression for details.

**GWAS catalog data integration**. GWAS catalog associations for GWAS of European ancestry were split into trait-specific data sets using experimental factor ontology terms. For every trait, a locus was defined as an associated variant $+/- 1$ Mb, and only loci that were replicated were retained. For proxy computation, we used our Egyptian cohort ($n = 110$) and the European individuals of 1000 Genomes ($n = 503$). For details, see Supplementary Methods: Data integration with the GWAS catalog.

**Integrative genomics view**. We implemented a workflow to extract all Egyptref data for view in the IGV[53]. This includes all sequencing data mapped to GRCh38 (cf. Supplementary Methods: Sequencing read mapping to GRCh38) as well as all assembly differences (cf. Supplementary Methods: Alignment to GRCh38 and Supplementary Methods: Assembly-based variant identification) and all Egyptian variant data. See Supplementary Methods: Gene-centric integrative data views for details.

**Reporting summary**. Further information on research design is available in the Nature Research Reporting Summary linked to this article.

## Data availability

All summary data of the Egyptian genome reference are available at www.egyptian-genome.org, where also variant allele frequencies can be queried online. Raw sequencing data and variant data are available at EGA under study ID EGAS00001004303 (Data set IDs: EGAD00001006034, EGAD00001006035, EGAD00001006036, EGAD00001006037, EGAD00001006038, EGAD00001006039, EGAD00001006040). De novo assemblies EGYPT, EGYPT_wtdbg2 and EGYPT_falcon have been deposited at DDBJ/ENA/GenBank under the accessions JABMBW000000000, JABMBX000000000, JABMBY000000000. The versions described in this paper are versions JABMBW010000000, JABMBX010000000 and JABMBY010000000, respectively. Assemblies of a Korean and a Yoruba genome have been obtained from GenBank under assembly accession IDs GCA_001750385.2 and GCA_001524155.4, respectively. Raw sequencing data of Pagani et al.[22] used in this study is available at EGA under data set IDs EGAD00001001372 and EGAD00001001380. World-wide genotype data used for population genetics analyses has been compiled from five sources, and accession codes and/or links are provided in Supplementary Data 12.

## Code availability

Github repositories lied_egypt_genome, Egyptref and 2020_population_genetics with workflows and scripts used in this study are available under the buschlab Github account at https://github.com/orgs/buschlab.

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

## Acknowledgements

We acknowledge support on coordination of the project and assembly work w.r.t the FALCON-based assembly through Ms. Lu Wang from the Novogene (UK) Company Limited. IW, HB and SI acknowledge funding by the Deutsche Forschungsgemeinschaft (DFG, German Research Foundation) under Germany's Excellence Strategy—EXC 22167-390884018. Verónica Calonga-Solís was supported by a scholarship from Deutscher Akademischer Austauschdienst (DAAD) and Coordenação de Aperfeiçoamento de Pessoal de Nível Superior (CAPES). All authors acknowledge computational support from the OMICS compute cluster at the University of Lübeck. Open access funding provided by Projekt DEAL.

## Author contributions

H.B., S.I., and M.S. conceived the study. I.W., A.K., M.M., H.B., and S.I. designed the study. I.W., A.K., M.M., M.O., A.F., and V.C.-S. performed data analysis. C.M. constructed the FALCON-based assembly. M.S. and S.E-M. compiled the Egyptian cohort and provided samples. M.H. performed mtDNA library preparation and sequencing. I.W., H.B., and S.I. wrote the manuscript. All authors read and approved the final manuscript.

## Competing interests

The authors declare no competing interests.
