## [Peer Review File · Nature Communications]

Reviewers' comments:

Reviewer #1 (Remarks to the Author):

The authors present a newly assembled Egyptian individual assembly, identified variants in comparison to the current human genome reference, and phased the variants using linked-reads. Furthermore, the authors identified allele frequency of each variants using short-read data of 100 individuals. The sequence data, newly assembled reference, and the population specific genotype allelic frequency will be useful for finding medically relevant variants in any type of genotype-phenotype related studies. However, there are some major and minor concerns that could significantly improve the manuscript and help better communicate with the audience.

1. Line 23-24; Authors claim variants are phased to maternal and paternal haplotypes. In Line 67, authors claim they generated a "phased de novo assembly". In fact, the authors generated a collapsed assembly and identified / phased small variants using 10X Genomics linked-reads using GRCh38 as the reference, not the assembly. Linked-reads enable local phasing on short variants. Without having the parental genotypes, it is not possible to identify the paternal / maternal haplotypes. This needs to be corrected. For claiming a 'phased de novo assembly', one would expect to see SINE/LINE repeats when comparing both haplotypes to each other, which is not shown in this manuscript. In fact, a fully phased assembly requires phasing large structural variations, including the sex chromosomes. I understand obtaining true haplotype is out of scope. It would be better to be focused on variants commonly shared among Egyptian population. This needs to be better communicated.

2. Is the assembled individual a male? Or a female? How are the sex chromosomes assembled? If the sex chromosomes weren't manually curated, it needs to be mentioned at least once in the main text that the analysis was only performed on the autosomes.

3. The authors chose WTDBG2-based assembly as the base line, "because it performs comparable or better according to various quality control (QC) measures". Which metrics did the authors consider specifically? The EGYPT_falcon assembly seems to have better k-mer based completeness. Continuity (NG50 and above) also seems better in the falcon version. The num. of misassemblies and k-mer misjoins could be an artifact caused by real structural differences compared to the reference (GRCh38).

4. When comparing the meta-assembly to GRCh38, were there any novel insertions / complex sequences found that are commonly shared among Egyptian population? All variant analysis is using GRCh38 as the reference. Not using the newly assembled Egyptian individual seems rather odd. Including variants that can be found only from the newly assembled genome - typically hard to call from short-reads only using GRCh38 as the reference - will improve the overall impact significantly of this manuscript.

5. Table 1. is showing initial results from QUASt-LG. Additional validations needs to be provided regarding the # misassemblies, as QAST-LG does not account for population/individual specific variations that could be counted as mis-assemblies when aligning an assembly that is structurally different from the reference. This will penalize genomes more divergent from the current GRCh38, where >70% of the GRCh38 is representing one individual from African-European ancestry (Schneider et al, 2017).

6. Additional assembly quality metrics will be useful; for example, showing k-mer multiplicity with KAT (Mapleson et al, 2017) will show overall k-mer completeness, missing sequences / extra sequences that needs to be removed. A comparable or better read depth distribution of Pacbio reads to the assembled individual vs. GRCh38 will be also a good indicator.

7. What is the base level QV of the new assembly? The illumina reads can be used for calculating the estimated QV.

Minor comments

1. Reference 8 and 9 are swapped

2. Supp. 3-7 needs improvements. Sort the GRCh38 by chromosome numbers and note in the label.

3. Line 104: 98.99% of variants being phased - out of how many variants called?
4. In Fig 4, this is showing a very heterozygous region. Show the phased variants in different tracks along with the phased block boundaries.

Reviewer #2 (Remarks to the Author):

It is interesting to read through this manuscript and the following positive points are made:

- De novo human genome assemblies are a rare resource; this project is even more eminent as it is from a region whose populations are poorly represented in global genome sequencing initiatives.
- Very good data quality sequenced at an overall coverage of about 270x.
- Robust comparison in terms of data quality with the published de novo Korean (AK1) and Yoruba genomes.
- Availability of phasing information, resulting in 98.99% of variants being phased.
- Overall it is a high-quality data from an under-represented population, and the analyses performed is extensive and in line with that of the published de novo genome sequenced data.

Suggestions for further analysis

The following suggestions are made for authors to address:

- Regional middle eastern population genome sequence data were not used for comparison, for example in the PCA analysis. Recently, there has been quite a few additions in terms of genome resources from the Middle Eastern populations in Kuwait, Qatar and the GME database. The GME reportedly has a number of Egyptian samples. However, the authors mentioned none of these or used these data which is closest to the Egyptians geographically.
- Analysis on Runs of Homozygosity and IBD regions is missing, taking into consideration the fact that rate of consanguinity is generally very high among the Arabs.

Reviewer #3 (Remarks to the Author):

The manuscript by Wohlers et al. presents a de novo assembly of human genome from an Egyptian individual with their descriptive parameters, plus short-read sequencing data of ten additional Egyptian individuals.

The manuscript is descriptive and, therefore, it is not clear the added value of providing a new de novo assembly of a human genome. The authors should stress what is the added value to the reference genome; how this new de novo assembly provides new information to the reference genome besides providing the description of new variants, which could be obtained by a resequencing process.

The sequencing of some Egyptian individuals is not an added value since recent whole genome data from Egyptians at decent coverage (~30X) is already available (Pagani et al 2015; ElHefnawi et al 2018; Serra-Vidal et al 2019). In fact, most of the analyses performed in the manuscript use the large dataset available from Pagani et al 2015.

As stated above, most of the analyses are descriptive, not performed in depth. An example of that is the European and African admixture approach based on a PC analysis and a description of mitochondrial lineages, which is very basic when dealing with whole-genome sequences that can provide more refined information (nonetheless, most of these refined analyses are already performed in the original paper from Pagani et al 2015 where most of the present data was already published). Another example is the tag-SNP analysis that is merely descriptive.

In sum, the authors should make an effort to explain the added value of a de novo assembly of a human genome and refine the analyses beyond the description of variants.

We thank the reviewers for their detailed comments and suggestions, which we are pleased to address below in the point-by-point response and in the revised manuscript. To facilitate re-review, all manuscript modifications in response to the reviewers' comments have been highlighted in blue. Other changes are highlighted in purple.

Reviewers' comments:

Reviewer #1 (Remarks to the Author):

The authors present a newly assembled Egyptian individual assembly, identified variants in comparison to the current human genome reference, and phased the variants using linked-reads. Furthermore, the authors identified allele frequency of each variants using short-read data of 100 individuals. The sequence data, newly assembled reference, and the population specific genotype allelic frequency will be useful for finding medically relevant variants in any type of genotype-phenotype related studies. However, there are some major and minor concerns that could significantly improve the manuscript and help better communicate with the audience.

1. Line 23-24; Authors claim variants are phased to maternal and paternal haplotypes. In Line 67, authors claim they generated a “phased de novo assembly”. In fact, the authors generated a collapsed assembly and identified / phased small variants using 10X Genomics linked-reads using GRCh38 as the reference, not the assembly. Linked-reads enable local phasing on short variants. Without having the parental genotypes, it is not possible to identify the paternal / maternal haplotypes. This needs to be corrected. For claiming a ‘phased de novo assembly’, one would expect to see SINE/LINE repeats when comparing both haplotypes to each other, which is not shown in this manuscript. In fact, a fully phased assembly requires phasing large structural variations, including the sex chromosomes. I understand obtaining true haplotype is out of scope. It would be better to be focused on variants commonly shared among Egyptian population. This needs to be better communicated.

Response: We fully agree with the reviewer's comments and apologize for using the misleading terms maternal/paternal haplotypes and phased assembly. As the reviewer states, we generated a collapsed assembly and performed short variant phasing for the same individual without considering parental information. We agree that this should have been made clearer, especially in light of recent advances towards obtaining fully phased assemblies. We rephrased the abstract (line 24-25) and the main text (“phased” removed on line 71, line 254) accordingly.

2. Is the assembled individual a male? Or a female? How are the sex chromosomes assembled? If the sex chromosomes weren't manually curated, it needs to be mentioned at least once in the main text that the analysis was only performed on the autosomes.

Response: The assembly individual is male. We added this information to the main text (line 81). Sex chromosomes have been assembled, but were not manually curated. This information is added to the main text (line 90).

3. The authors chose WTDBG2-based assembly as the base line, “because it performs comparable or better according to various quality control (QC) measures”. Which metrics did the authors consider specifically? The EGYPT_falcon assembly seems to have better k-mer based completeness. Continuity (NG50 and above) also seems better in the

falcon version. The num. of misassemblies and k-mer misjoins could be an artifact caused by real structural differences compared to the reference (GRCh38).

Response: We agree with the reviewer's assessment of the FALCON-based versus the WTDBG2-based assembly. Neither algorithm outperforms the other one in all aspects. Nevertheless, we found that the FALCON-based assembly had about three times more misassemblies than the WTDBG2-based at similar contingency and length, which ruled out structural differences as an alternative explanation. This result was confirmed by the classification of misassemblies (also to answer point 5 of reviewer #1). The FALCON-based assembly has more misassemblies overall, as well as outside regions that are difficult to assemble such as segmental duplication regions or centromeres. This still holds when removing possible SVs from the misassemblies. At the same time, the WTDBG2-based assembly is comparable to other genome assemblies even without scaffolding. There are no N bases and thus no scaffolding errors, which is another reason why we favored WTDBG2. We justified our choice for the WTDBG2-based assembly in the main manuscript on lines 90-96, which read:

The WTDBG2-based assembly was selected as the meta assembly basis, because it performs similarly or better than the FALCON-based assembly, according to various quality control (QC) measures. The former did not require scaffolding, and thus there are no N bases or scaffolding errors. Overall, it has about 50% fewer misassemblies. This QC measure holds true even when ignoring misassemblies in centromeres and in segmental duplications and after correction for structural variants.

We rewrote the respective paragraph to make clear that both base assemblies are complementary and of high quality.

Both the base assemblies and the final meta assembly are of high quality and complementary and they are comparable to the publicly available assemblies of a Korean 2 and a Yoruba (GenBank assembly accession GCA_001524155.4) individual in terms of genome length and various quality control (QC) measures, (Table 1, extended version in Suppl. Table 2). Assembly quality is confirmed by quality control (QC) measures assessed by QUAST-LG³³ (Suppl. Table 2), NA-values (Suppl. Fig. 2), k-mer multiplicity with KAT³⁴ (Suppl. Fig. 3, 4 and 5), QV values of more than 40 and by dot plots of alignment with reference GRCh38 (Suppl. Figs. 6-10).

Both base assemblies will be publicly available along with the final meta assembly (Section Data availability), which we now mention explicitly.

4. When comparing the meta-assembly to GRCh38, were there any novel insertions / complex sequences found that are commonly shared among Egyptian population? All variant analysis is using GRCh38 as the reference. Not using the newly assembled Egyptian individual seems rather odd. Including variants that can be found only from the newly assembled genome - typically hard to call from short-reads only using GRCh38 as the reference - will improve the overall impact significantly of this manuscript.

Response:

This is a very good question and remark. To address this point, which is also made by reviewer #3, we searched for novel sequences in our assembly that are absent from the GRCh38 reference. The corresponding paragraph now added to the main text reads:

Using the EGYPT *de novo* assembly, we searched for unique insertions that are common in Egyptians. Towards this, we first mapped all short-read data against the GRCh38 reference genome and to other decoy or alternative haplotype sequences from the GATK bundle. All reads that could not be mapped were subsequently mapped against the EGYPT *de novo* assembly. A similar approach was recently applied to identify novel, unique insertions in *de novo* assemblies of 17 individuals from 5 populations using 10x genomics sequencing³⁶. Altogether we identified 40 unique insertions longer than 500 bp with a total length of 40kb, for which we required for every base in the identified region to have a minimal coverage of 5 reads in at least 10 Egyptian individuals (Suppl. Table 9). Of these sequences, 28 have been mentioned before by Wong *et al.*³⁶, and 10 more in different studies within the last 15 years^{37 38 39 40}. Two out of the 40 insertions are most likely novel. In addition, one region contains three unique insertions, of which two contain additional, novel sequences longer than 500 bases. Closer inspection reveals that these sequences are located within a region of two 50 kb gaps (i.e. N sequences) in the GRCh38 reference genome at chromosome loci chr13:111,703,856-111,753,855 and chr13:111,793,442-111,843,441 with about 40 kb of reference sequence between the gaps. The EGYPT, AK1 and YORUBA assembly sequences that cover this 140 kb reference sequence from chr13:111,703,856 to 111,843,441 are very similar (Suppl. Figs. 23, 24 and 25). They all align about 4 kb from the 40 kb reference sequence between the gaps, only, but at the very beginning of the respective assembly sequence (Suppl. Figs. 26, 27 and 28). Performing a BLAST search of the 140 kb EGYPT assembly sequence reveals an overall 44 kb alignment in five, mainly consecutive, large alignment blocks to “Homo sapiens chromosome 13 clone WI2-2182D8” (AC188786.1) from position 1 to 44,382, see Suppl. Fig. 29. This large reference genome region that contains the largest gap covering sequence reported for AK1² is not resolved yet.

Overall, we identified 330 single nucleotide variants and indels in 36 of 40 non-reference sequences (Suppl. Table 10). The percentage of reads that could not be mapped to GRCh38 or GATK bundle sequences, but which were mappable against the *de novo* assembly is on average 8.6%, but for some individuals up to 34.2% (cv. Suppl. Fig. 30). Previously unmapped short reads of 110 Egyptians covered positions for more than 19 Mb of the Egyptian *de novo* assembly. Unique sequences that are commonly shared among Egyptians illustrate that additional reference genomes are needed to capture the genetic diversity that are neither assessable by short read sequencing nor with the current human reference genome.

In addition, the large number of assembly positions to which such short reads map which could not be mapped to the reference genome GRCh38 (including widely used supplementary sequences included in the GATK bundle), indicate a need for further assembly-based reference data and for new approaches to better capture genetic diversity.

This corresponding section was added to the Methods in the main manuscript:

Unique inserted sequences

We trimmed Illumina short sequencing reads of 110 Egyptian individuals using FASTP 0.20.0 with default parameters, mapped the output reads to GRCh38 and GATK bundle sequences using BWA 0.7.15-r1140 and sorted by chromosomal position using SAMTOOLS 1.3.1. Subsequently, we extracted reads that did not map to GRCh38 using SAMTOOLS with parameter F13 (i.e. read paired, read unmapped, mate unmapped) and repeated the mapping and sorting using the Egyptian *de novo* assembly. We merged the read-group specific BAM files for each sample and calculated the per base read depth using SAMTOOLS. Afterwards, we aggregated the results via custom scripts and extracted uniquely inserted sequences from the Egyptian *de novo* assembly. Insertions were defined as contiguous regions of at least 500 bp having a coverage of more than 5 reads per base in 10 or more samples. Lastly, we BLASTed the obtained sequences against the standard databases (option nt) for highly similar sequences

(option megablast) using a custom script. For the uniquely inserted sequences that we identified, we created a pileup over all BAM files containing the reads that did not map to GRCh38 using SAMTOOLS. Based on these pileups, we then called the variants using BCFTOOLS. Variants with quality of more than 10 were kept.

5. Table 1. is showing initial results from QUASt-LG. Additional validations needs to be provided regarding the # misassemblies, as QAST-LG does not account for population/individual specific variations that could be counted as mis-assemblies when aligning an assembly that is structurally different from the reference. This will penalize genomes more divergent from the current GRCh38, where >70% of the GRCh38 is representing one individual from African-European ancestry (Schneider et al, 2017).

Response:

We addressed this point by using (and slightly adapting) a workflow that was recently published for correcting QUASt-LG misassembly numbers for SVs and in centromeric regions or segmental duplications:

Shafin *et al.* Efficient *de novo* assembly of eleven human genomes using PromethION sequencing and a novel nanopore toolkit. bioRxiv 715722

The following text was added to the main section:

The WTDBG2-based assembly was selected as the meta assembly basis, because it performs similarly or better than the FALCON-based assembly, according to various quality control (QC) measures. The former did not require scaffolding, and thus there are no N bases or scaffolding errors. Overall, it has about 50% fewer misassemblies. This QC measure holds true even when ignoring misassemblies in centromeres and in segmental duplications and after correction for structural variants.

This text was added to the main Methods section:

Additionally, we removed misassemblies in centromeres or in segmental duplication regions from the QUASt-LG report and furthermore removed structural variants from misassemblies

The following text was added to the Suppl. Methods:

We used an approach utilized in two recent assembly publications (Shafin *et al.*, 2019)(Nurk *et al.*, 2020) to better characterize QUASt-LG misassemblies and to correct for structural variants. We adapted the script of Shafin *et al.* from https://github.com/kishwarshafin/helen/blob/master/helen/modules/python/helper/quast_sv_extractor.py, together with the centromere and segmental duplication files that the authors used, which were downloaded from <https://www.ncbi.nlm.nih.gov/grc/human> and <https://github.com/mvollger/segDupPlots/>, respectively. Structural variants that we removed were obtained from the DELLY short read-based SV calls for the assembly individual. For the AK1 assembly, we obtained structural variants from Suppl. Table 16 of Seo *et al.*, 2016. Resulting stratified misassembly numbers for all assemblies are provided together with the QUASt-LG QC numbers in Suppl. Table 2.

With respect to point 3 of Reviewer #1, we conclude that the WTDBG2-based assembly has significantly lower number of misassemblies compared with the FALCON-based assembly, also when disregarding SVs and/or other regions that are challenging to assemble. This confirms the main reason for choosing the WTDBG2-based assembly as base assembly.

6. Additional assembly quality metrics will be useful; for example, showing k-mer multiplicity with KAT (Mapleson et al, 2017) will show overall k-mer completeness, missing sequences / extra sequences that needs to be removed. A comparable or better read depth distribution of Pacbio reads to the assembled individual vs. GRCh38 will be also a good indicator.

Response:

As suggested, we assessed k-mer multiplicity via KAT, and confirmed k-mer completeness in the two draft and the final assemblies without missing or extra sequences. In line with the QUAST-LG duplication ratio of 1.01, KAT indicates a very slightly higher ratio for the meta assembly, which is likely caused by integrating FALCON assembly sequences in the WTDBG2 draft assembly. The KAT result figures have been added to the Supplement (Suppl. Figs. 3, 4 and 5) and the following sentence was added to the Main section

Assembly quality is confirmed by quality control (QC) measures assessed by QUAST-LG³³ (Suppl. Table 2), NA-values (Suppl. Fig. 2), k-mer multiplicity with KAT³⁴ (Suppl. Fig. 3, 4 and 5), QV values of more than 40 and by dot plots of alignment with reference GRCh38 (Suppl. Figs. 6-10).

To the Main methods section we added the sentence

K-mer multiplicity was assessed with KAT³⁴.

7. What is the base level QV of the new assembly? The illumina reads can be used for calculating the estimated QV.

Response:

As suggested, we used the Illumina reads to compute the QV, following the approach described in

Porubsky *et al.* A fully phased accurate assembly of an individual human genome. bioRxiv 855049

We added the QV values and corresponding information for its computation for both draft assemblies and final meta assembly to Suppl. Table 2, as such:

	EGYPT	EGYPT_wtdbg2	EGYPT_falcon
Base level QV			
Sum base differences called	162418	140965	144598
Effective genome size	2820489739	2733934177	2897551797
QV	42.4	42.9	43.0

We added the info to the main section as

Assembly quality is confirmed by quality control (QC) measures assessed by QUAST-LG³³ (Suppl. Table 2), NA-values (Suppl. Fig. 2), k-mer multiplicity with KAT³⁴ (Suppl. Fig. 3, 4

and 5), QV values of more than 40 and by dot plots of alignment with reference GRCh38 (Suppl. Figs. 6-10).

and to the Methods section as

Following Porubsky *et al.*⁶⁵, we computed QV as the number of homozygous variants divided by the effective genome size. Towards this, we mapped all short reads to the assembly using BWA MEM and perform variant calling using FREEBAYES with default parameters. We kept only homozygous variants with a minimum quality of 10 using VCFTOOLS. Single-nucleotide differences were counted as difference of 1 bp, indel differences as the length differences between reference and alternative allele. Based on SAMTOOLS command “stats”, we computed the sum of bases with short read coverage as effective genome size.

Minor comments

1. Reference 8 and 9 are swapped

Response: We corrected the swapped references 8 and 9.

2. Supp. 3-7 needs improvements. Sort the GRCh38 by chromosome numbers and note in the label.

Response: We adjusted the tables accordingly.

3. Line 104: 98.99% of variants being phased - out of how many variants called?

Response:

We added this information to the main section. We realized that for the computation of this percentage, the 10x Genomics pipeline uses also homozygous reference variants from the cohort-based variant calling. We now provide the number and percentage of phased variants excl. homozygous reference positions in the main manuscript. To avoid confusion, we added an explanation in the label of Suppl. Table 5, which shows the summary report of the 10x Genomics pipeline. The adjusted sentence reads:

This resulted in 3,834,900 of 4,008,080 autosomal variants being phased (95.7%).

4. In Fig 4, this is showing a very heterozygous region. Show the phased variants in different tracks along with the phased block boundaries.

Response:

We generated the suggested graphical view using the 10x Genomics visualization Software LOUPE and added it to the Supplement (Suppl. Fig. 48), referring to it in the caption of Fig. 4. The phase block in which the BRCA2 gene lies is about 8 Mb (from chr13:25,831,216-33,523,430). However, when zooming out to see phase block boundaries, variants are not displayed anymore. Thus, we don't display the phase block boundaries.

In the main Section, we added this info as

As an example for personalized medicine for Egyptian specific genetics we visualize the complete genetic information of the DNA repair-associated gene BRCA2 from our study in the integrative genomics viewer⁵⁷ (IGV) and the variant phasing information within the 10x Genomics browser LOUPE in Fig. 4 and Suppl. Fig 48, respectively.

Reviewer #2 (Remarks to the Author):

It is interesting to read through this manuscript and the following positive points are made:

- **De novo human genome assemblies are a rare resource; this project is even more eminent as it is from a region whose populations are poorly represented in global genome sequencing initiatives.**
- **Very good data quality sequenced at an overall coverage of about 270x.**
- **Robust comparison in terms of data quality with the published de novo Korean (AK1) and Yoruba genomes.**
- **Availability of phasing information, resulting in 98.99% of variants being phased.**
- **Overall it is a high-quality data from an under-represented population, and the analyses performed is extensive and in line with that of the published de novo genome sequenced data.**

Suggestions for further analysis

The following suggestions are made for authors to address:

- **Regional middle eastern population genome sequence data were not used for comparison, for example in the PCA analysis. Recently, there has been quite a few additions in terms of genome resources from the Middle Eastern populations in Kuwait, Qatar and the GME database. The GME reportedly has a number of Egyptian samples. However, the authors mentioned none of these or used these data which is closest to the Egyptians geographically.**

Response:

We fully agree with reviewer #2 that it is very interesting to resolve Egyptian admixture on a finer scale with respect to geographically close populations. So far, however, we had restricted our population genetic analysis to whole genome sequencing-based data. Until last year the 1000 genomes data set was the only available source, which, however, does not contain North African or Middle Eastern populations or individuals.

For the revision, we thus included additional population genetic analyses for world-wide populations, comprising the currently largest number of diverse populations. Towards this, we included a very recent paper (March 2020) which published whole genome sequencing-based variant data covering 929 individuals from 54 diverse human populations (Bergström *et al.*, 2020). This allowed us to add variant data from geographically close populations, which are either based on SNP arrays (Fernandes *et al.*, 2019) or whole exome sequencing (Kuwait, Qatar, GME). In fact, we obtained the Greater Middle East (GME) data set, the most comprehensive of the region, from dbGAP and performed admixture analysis. Unfortunately, the data download from dbGAP did not include detailed enough population annotation (e.g. samples are annotated with “Arabic”, only). We contacted the corresponding author on January 30 and again March 11 asking for the annotation that was used in the GME paper, but have not received a reply so far and had to leave out this data set for now. Alternatively, we obtained the largest number of geographically close populations by combining all available whole genome data

(1000G; Bergström *et al.* 2020; 108 Qatari from Rodriguez-Flores *et al.*, 2016 and 110 Egyptians) with recent SNP array data from the region (Dubai, Iran, Oman, Saudi Arabia, Yemen) and from 68 additional populations (Europe, Africa, Western Asia and South Asia). The array data was published by Fernandes *et al.*, 2019 and Busby, 2020, respectively. For details see Suppl. Table 12. We now also included an admixture analysis, which identifies world-wide genetic components that occur in Egyptians.

The summary of this more sophisticated population genetic analysis replaces in the main section the previous population genetics paragraph that used only 1000G data and reads as follows:

We genetically characterized the Egyptian population with respect to 143 other populations of the world using variant data of 5,429 individuals in total. For this, we combined five different data sets: (1) a recently published whole genome sequencing (WGS)-based variant data set from 929 individuals of the Human Genome Diversity Project (HGDP), covering 51 populations⁴¹; (2) 2,504 individuals from 26 populations of the 1000 Genomes project for which phase 3 genotypes are available⁵; (3) WGS-based variant data from 108 Qatari individuals⁴²; (4) SNP array-based variant data of 478 individuals from five countries of the Arabian Peninsula²⁵; (5) 1,305 individuals from 68 African, European, Western and Southern Asian populations that were compiled from 8 different publications into a recent SNP array-based variant data set⁴³. All individuals and their annotations are provided in Suppl. Table 11, data sources are described in Suppl. Table 12. A principal component analysis of the data shows a genetic continuum between Europeans, Africans, East Asians and Americans along the first three principal components, see Suppl. interactive HTML-based Fig. PCA_interactive.html. Egyptians are located on the European-African axis and close to Europeans. Their genetic variance spreads to a small degree in the direction of the Asian axis, akin to further individuals from the Middle East (see Fig. 2c). To preclude a technical bias when intersecting WGS with SNP array data, we compared the analysis results when using whole genome data, only, or when intersecting WGS data with SNP arrays and found comparable results in both cases (Suppl. Fig. 38). The Egyptian PCA location is further supported by an admixture analysis. Our analysis specifies $k=24$ as the optimal number of genetic components for the entire data set, i.e. having the smallest cross validation error (see Suppl. Fig. 39 for results for $k=10$ to $k=25$). Accordingly, the genetics of Egyptian individuals comprises four distinct population components that sum up to 75% on average. Egyptians have a Middle Eastern, a European / Eurasian, a North African and an East African component with 27%, 24%, 15% and 9% relative influence, respectively (see Fig. 2a). According to our cohort, Egyptians show genetically little heterogeneity, with little variance in the proportion of individual components between the individuals (Suppl. Figs 40 and 41). With a focus on populations from the Horn of Africa, the four components we identified have been described before by Hodgeson *et al.*⁴⁴ in a cohort of 2,194 individuals from 81 populations (mainly 1000 Genomes and HGDP) and substantially fewer variants ($n=16,766$), but including also 31 Egyptians. They and others hypothesize that most non-African ancestry, i.e. the Eurasian / European and Middle Eastern components in the populations from North Africa and the Horn of Africa is resulting from prehistoric back-to-Africa migration^{44 24}. Recently, Serra-Vidal *et al.* describe North Africa as a “melting pot of genetic components”, attributing most genetic variation in the region also to prehistoric times⁴⁵. Here, we confirm previously identified genetic components, yet using 2.5 times as many individuals, and using WGS data for the majority of them. This is thus the hitherto most comprehensive data set on genetic diversity world-wide and in this region.

The corresponding Methods section has been changed to

Population genetics

For population genetic analyses, we compared the Egyptian variant data with variant data from five additional sources specified in Suppl. Table 12. Individuals together with their annotations are listed in Suppl. Table 11. Variant data was merged to contain only variants present in all data sets and subsequently filtered and LD pruned. Genotype principal component analysis was computed using SMARTPCA⁷⁵ from the EIGENSOFT package. Admixture was computed with ADMIXTURE⁷⁶ (cf. Suppl. Methods: *Population genetics and SNP array-based Egyptian variant data*).

Suppl. Methods contain additional details on the analyses. We slightly changed the paragraph on mitochondrial haplogroups, now stating that these support our admixture results.

• Analysis on Runs of Homozygosity and IBD regions is missing, taking into consideration the fact that rate of consanguinity is generally very high among the Arabs.

Response:

We added a figure comparing overall length of ROHs for different world populations to the updated Figure 2, to comprehensively genetically characterize the Egyptian population also with respect to consanguinity and IBD. We added after the updated and extended paragraph concerning Egyptian population genetics these sentences with respect to runs of homozygosity and IBD

Lastly, we characterized the Egyptian population with respect to runs of homozygosity. The distribution of overall length of ROHs larger than 5 Mb is comparable for the Egyptian population and Middle Eastern populations and, to lesser extent, also for other North African and Western Asian populations. In comparison, Europeans and Sub-Saharan Africans have usually shorter ROHs, see Fig. 2b. Abundance of long ROHs is typical for the Greater Middle East²⁶ and reflects the common practice of consanguineous marriages in this region.

We further added a Suppl. Table with ROH regions of the assembly individual, as these are of special interest, and summarize in the main manuscript:

We identified 22 runs of homozygosity (ROH) (Suppl. Table 6), out of which 16 are larger than 5 Mb and sum up to overall 192 MB, which indicates consanguinity at the level of parental third-degree relationship³⁵.

The main Methods have been updated with these sentences within the novel Section “Population Genetics”

Runs of homozygosity were computed on the same files that were used for PC computation and admixture using PLINK `-homozyg`. ROHs with size larger than 5 Mb were summed to obtain overall length of ROHs per individual.

Reviewer #3 (Remarks to the Author):

The manuscript by Wohlers et al. presents a de novo assembly of human genome from an Egyptian individual with their descriptive parameters, plus short-read sequencing data of ten additional Egyptian individuals.

Response:

This is correct, but constitutes only a small subset of data generated and/or utilized and of analyses performed

- 1) Data generated in this study: (i) Personal Genome / *de novo* assembly: Overall 270x coverage of PacBio, 10X genomics, and Illumina short-read sequencing; individual-matching RNA-Seq (ii) Population genome: 10 whole genomes sequenced at high coverage (30x); mitochondrial sequencing of 227 individuals
- 2) Data used: 97 Egyptians sequenced at low coverage (7x) and 3 at high coverage (30x) by Pagani *et al.*; 398 Egyptians genotyped with SNP arrays (in-house, Bejaoui *et al.*, 2019)
- 3) Analyses performed: *de novo* assembly, calling of variants for all 110 Egyptians starting from raw data; mitochondrial haplotypes; allelic expression; population genetics; population-specific variants; novel variants; disease-related variants; possible impact of differences in linkage disequilibrium on genetic association studies

The manuscript is descriptive and, therefore, it is not clear the added value of providing a new *de novo* assembly of a human genome. The authors should stress what is the added value to the reference genome; how this new *de novo* assembly provides new information to the reference genome besides providing the description of new variants, which could be obtained by a resequencing process.

Response:

Also reviewer #1 mentioned that current variant analyses use GRCh38 as a reference and that **“Including variants that can be found only from the newly assembled genome - typically hard to call from short-reads only using GRCh38 as the reference - will improve the overall impact significantly of this manuscript.”** To address this common and valid point, we performed additional analyses to characterize and identify genetic variation that cannot be obtained from GRCh38 or a resequencing process. Please refer to our answer to remark 4 of reviewer #1.

The sequencing of some Egyptian individuals is not an added value since recent whole genome data from Egyptians at decent coverage (~30X) is already available (Pagani et al 2015; ElHefnawi et al 2018; Serra-Vidal et al 2019).

Response:

Please note, that the data set of Pagani *et al.*, which contributes 100 of the 110 individuals used in our study is sequenced at target genome coverage of 7x, and only 3 Egyptian individuals have been sequenced at 30x by Pagani *et al.* We add 9 Egyptian individuals sequenced at 30x, and one at 90x. In terms of sequence coverage, we add 360x to the about 770x of Pagani *et al.* We re-compute all variants in this combined cohort using joint genotyping, and adding our high coverage samples is important to improve the variant calling. ElHefnawi published about variants from only two short read, whole genome sequenced Egyptians. Serra-Vidal *et al.* published their paper “Heterogeneity in Palaeolithic Population Continuity and Neolithic Expansion in North Africa” in November 2019, after the submission of our manuscript; their study includes 2 WGS Egyptians and few SNP array-based Egyptians from the Human Origins data set, which are also contained in our admixture analysis. In the revised manuscript, we cite them and state that they attribute most genetic variation in the region to prehistoric times. In summary, to the best of our knowledge, the data set we compiled constitutes the most comprehensive Egyptian whole genome sequencing-based variant data to date.

In fact, most of the analyses performed in the manuscript use the large dataset available from Pagani et al 2015.

Response:

We respectfully disagree. The paper of Pagani *et al.* is titled “Tracing the Route of Modern Humans out of Africa by Using 225 Human Genome Sequences from Ethiopians and Egyptians” and it relates to the field archaeogenetics. The only type of analysis that the Pagani *et al.* and our paper share is the genetic characterization of the Egyptians with respect to other populations using principal component analysis. For this we chose all African and European populations individuals from 1000G in the first version of the paper and a many more individuals and data sets in the revised version, while Pagani *et al.* selected 1000G and further population data they deemed relevant to trace the route of modern humans out of Africa.

As stated above, most of the analyses are descriptive, not performed in depth. An example of that is the European and African admixture approach based on a PC analysis and a description of mitochondrial lineages, which is very basic when dealing with whole-genome sequences that can provide more refined information (nonetheless, most of these refined analyses are already performed in the original paper from Pagani et al 2015 where most of the present data was already published).

Another example is the tag-SNP analysis that is merely descriptive.

Response:

We believe that our descriptive analyses of a population for which very little genetic information has been assessed so far form the basis of many ensuing in-depth analyses. This is particularly true when using the latest sequencing and assembly techniques together with novel genome analysis methods. We provide the data used for the descriptive summaries within Supplementary Tables, which can be the starting point for many in depth analysis. For example, the data underlying the tag SNP analysis is provided in Suppl. Table 17 and lists for every tag SNP the associated GWAS catalog data including the linked diseases, and we discuss Alzheimer’s disease SNP rs2075650 as example. We deliberately omitted lengthy discussions of individual findings within the main manuscript, because we believe it is beyond the scope of a paper that introduces a comprehensive Egyptian genome reference for the first time.

With respect to reviewer #3’s impression that particularly the population genetics analysis is not performed with sufficient depth, this analysis is now significantly extended and includes geographically closer populations, as suggested by reviewer #2. Additionally, we used SNP array-based variant data from 398 additional Egyptian individuals to replicate population genetics analyses using a different cohort and different set of variants. For details on the updated population genetics analysis, please refer to our answers for reviewer #2.

In sum, the authors should make an effort to explain the added value of a de novo assembly of a human genome and refine the analyses beyond the description of variants.

Response:

We addressed this point with two extensive additional analyses: (1) by targeting genetic variation that can be obtained from the *de novo* assembly, but not from the reference genome GRCh38. (2) by a comprehensive and more sophisticated population genetics analysis comprising more than 4000 individuals.

REVIEWERS' COMMENTS:

Reviewer #1 (Remarks to the Author):

The authors addressed all my raised concerns very well. I see the manuscript has substantially gained novelty and has improved on various aspects. I especially enjoyed reading the section describing novel variants unique to EGYPT compared to GRCh38, as well as the interactive 3d-PCA. Still, I find a few sentences that could be shortened or rephrased, including a few that could be better removed for clarity. All my comments below are considered minor.

1. The first sentence of the abstract; "The human genome is composed of chromosomal DNA sequences consisting of bases A, C, G and T – the blueprint to implement the molecular functions that are the basis of every individual's life." I don't think this needs to be highlighted as the first sentence in the abstract. Please remove.

2. In Table 1, number of lines contain information not necessarily needed or add more confusion. The authors display all metrics from QUASt LG, which some cannot be applied in this comparison. Under mismatches, # mismatches per 100kbp / # indels per 100kbp and stats under the K-mer based statistics are all based on the reference. These statistics are no longer accurate when the assembly of comparison has a similar or better continuity in certain regions. # mismatches will be affected by the variants not included in the reference. # misassemblies also fall under this category, if it is just a naïve comparison between the reference and the assembly. If this was measured without the use of GRCh38, add an explanation to the caption.

I'd suggest to remove all unnecessary statistics and show only those important to emphasize the quality of assemblies, between EGYPT assemblies or across AK1 and YORUBA.

A good example assembly metric is described in Rhie et al.,
<https://doi.org/10.1101/2020.05.22.110833> .

3. Page 4 line 93-94, "Overall, it has about 50% fewer misassemblies." This should be removed or rephrase "misassemblies" to "disagreements to the reference" as done in the Shafin et al. paper. This "misassembly" measure in QUASt-LG is an approximation given a reference, and there is always a chance of a true individual-specific variant captured. This is also true for the Shafin et al. paper, and thus they do one more round of comparison against the highly curated T2T chromosome X assembly of the same individual CHM13 (Miga et al., <https://doi.org/10.1101/735928>). To find true mis-assemblies, an orthogonal validation needs to be applied (coverage analysis and/or information from long-range sequencing data etc.). However, I understand this is a bit out of scope at the stage of justifying why the WTDBG-based assembly was chosen. Continuity is in fact an important metric, and it does not hurt the overall message of this paper to remove or rephrase this sentence.

4. Page 5 line 107, "We performed repeat annotation and repeat masking for all assemblies": this is a boring sentence. Provide some more results, for example: "When annotating repeats with RepeatMasker, ~49% of the genome were identified as repetitive with its contents similarly identified as in AK1 or YORUBA (Supplementary Table 4)".

5. Page 6 line 143-144, "A similar approach was recently applied to identify novel, unique insertions in de novo assemblies of 17 individuals from 5 populations using 10x genomics sequencing" could be shortened to the sentence in line 140: "Towards this, we took a similar approach as in Wong et al. (36) and mapped all short-read data against the GRCh38 reference genome including other decoy or alternative haplotype sequences using the GATK bundle."

6. Fig. 4, add to the caption that the shown region was included in a larger phased-block chr13:25,831,216-33,523,430.

7. Add what is shown to the X and Y axis of the dot plots (Supp. Fig. 23-28). For example, add

legend "GRCh38" or "EGYPT" to X and Y axis.

Arang Rhie

Reviewer #2 (Remarks to the Author):

The authors have suitably addressed our suggestions.

Please find our replies to the reviewers' comments below, they are highlighted blue.

REVIEWERS' COMMENTS:

Reviewer #1 (Remarks to the Author):

The authors addressed all my raised concerns very well. I see the manuscript has substantially gained novelty and has improved on various aspects. I especially enjoyed reading the section describing novel variants unique to EGYPT compared to GRCh38, as well as the interactive 3d-PCA. Still, I find a few sentences that could be shortened or rephrased, including a few that could be better removed for clarity. All my comments below are considered minor.

1. The first sentence of the abstract; “The human genome is composed of chromosomal DNA sequences consisting of bases A, C, G and T – the blueprint to implement the molecular functions that are the basis of every individual’s life.” I don’t think this needs to be highlighted as the first sentence in the abstract. Please remove.

This sentence is already removed in the suggested abstract of the editor, which we adopted with minor adjustments (mainly shortening).

2. In Table 1, number of lines contain information not necessarily needed or add more confusion. The authors display all metrics from QUASt LG, which some cannot be applied in this comparison.

Under mismatches, # mismatches per 100kbp / # indels per 100kbp and stats under the K-mer based statistics are all based on the reference. These statistics are no longer accurate when the assembly of comparison has a similar or better continuity in certain regions. # mismatches will be affected by the variants not included in the reference. # misassemblies also fall under this category, if it is just a naïve comparison between the reference and the assembly. If this was measured without the use of GRCh38, add an explanation to the caption.

I’d suggest to remove all unnecessary statistics and show only those important to emphasize the quality of assemblies, between EGYPT assemblies or across AK1 and YORUBA.

A good example assembly metric is described in Rhie et al.,
<https://doi.org/10.1101/2020.05.22.110833> .

We agree with the limitations of reference-based QC measures reported by QUASt-LG. The metrics suggested in the mentioned preprint are in fact better suitable to measure assembly quality. Thus, we moved those from Supplementary to main table, which were not listed in the main table yet. We think that the reference-based metrics are still characterizing the assemblies well (e.g. because of the exceptional quality of the human reference genome GRCh38), although they have the limitations noted by the reviewer. We thus now refer in the main Table to relevant metrics as “differences from GRCh38” instead of “mistakes”. Further, now all GRCh38-based metrics are denoted as such explicitly. Accordingly, we updated the legend of the table to denote a “Comparison of main assembly characteristics and quality metrics.” and omit denoting “best” assembly metrics in bold.”

3. Page 4 line 93-94, “Overall, it has about 50% fewer misassemblies.” This should be removed or rephrase “misassemblies” to “disagreements to the reference” as done in the Shafin et al. paper. This “misassembly” measure in QUASt-LG is an approximation given a reference, and there is always a chance of a true individual-specific variant captured. This is also true for the Shafin et al. paper, and thus they do one more round of comparison against the highly curated T2T chromosome X assembly of the same individual CHM13 (Miga et al., <https://doi.org/10.1101/735928>). To find true mis-assemblies, an orthogonal validation needs to be applied (coverage analysis and/or information from long-range sequencing data etc.). However, I understand this is a bit out of scope at the stage of justifying why the WTDBG-based assembly was chosen. Continuity is in fact an important metric, and it does not hurt the overall message of this paper to remove or rephrase this sentence.

As suggested, we rephrased “misassemblies” to “disagreements to the reference”.

4. Page 5 line 107, “We performed repeat annotation and repeat masking for all assemblies”: this is a boring sentence. Provide some more results, for example: “When annotating repeats with RepeatMasker, ~49% of the genome were identified as repetitive with its contents similarly identified as in AK1 or YORUBA (Supplementary Table 4)”.

We changed the sentence as suggested.

5. Page 6 line 143-144, “A similar approach was recently applied to identify novel, unique insertions in de novo assemblies of 17 individuals from 5 populations using 10x genomics sequencing” could be shortened to the sentence in line 140: “Towards this, we took a similar approach as in Wong et al. (36) and mapped all short-read data against the GRCh38 reference genome including other decoy or alternative haplotype sequences using the GATK bundle.”

We shortened this sentence as suggested.

6. Fig. 4, add to the caption that the shown region was included in a larger phased-block chr13:25,831,216-33,523,430.

We added this information to Fig. 4 as suggested.

7. Add what is shown to the X and Y axis of the dot plots (Supp. Fig. 23-28). For example, add legend “GRCh38” or “EGYPT” to X and Y axis.

We added the additional legends to the axes of Suppl. Fig. 23-28 as suggested.

Arang Rhie

Reviewer #2 (Remarks to the Author):

The authors have suitably addressed our suggestions.